# A Feasibility Study of the My Strengths Training for Life™ (MST4Life™) Program for Young People Experiencing Homelessness

**DOI:** 10.3390/ijerph19063320

**Published:** 2022-03-11

**Authors:** Jennifer Cumming, Fiona J. Clarke, Mark J. G. Holland, Benjamin J. Parry, Mary L. Quinton, Sam J. Cooley

**Affiliations:** 1School of Sport, Exercise and Rehabilitation Sciences, University of Birmingham, Birmingham B15 2TT, UK; f.j.clarke@bham.ac.uk (F.J.C.); bparry@clemson.edu (B.J.P.); m.quinton@bham.ac.uk (M.L.Q.); 2Institute for Mental Health, University of Birmingham, Birmingham B15 2TT, UK; 3Department of Health and Sport, Newman University, Birmingham B32 3NT, UK; mark.holland@newman.ac.uk; 4School of Psychology, University of Leicester, Leicester LE1 7RH, UK; sc747@leicester.ac.uk

**Keywords:** co-design, community engagement, intervention development, positive youth development, social inequalities, strengths-based

## Abstract

My Strengths Training for Life™ (MST4Life™) is a positive youth development program for improving wellbeing and social inclusion in young people experiencing homelessness. MST4Life™ addresses a gap in strengths-based programs aimed at promoting healthy and optimal development in vulnerable older adolescents/emerging adults. The program was co-developed with a UK housing service as part of a long-term (>8 years) community–academic partnership. This mixed-methods study describes a key step in developing and evaluating the program: exploring its feasibility and acceptability with 15 homeless young people (Mean age = 19.99 years, SD = 2.42; 60% male, 40% female). Participants experienced 8 weekly sessions within their local community, followed by a 4-day/3-night residential outdoor adventure trip. In addition to their attendance records, the viewpoints of the participants and their support workers were obtained using diary rooms and focus groups. Feasibility was indicated via the themes of attendance, engagement, and reaction. The findings suggested that young people enjoyed and perceived a need for the program, that they considered the program and its evaluation methods to be acceptable, and that both the community-based and outdoor adventure residential phases could be implemented as planned. Minor modifications are needed to recruitment strategies before it is more widely rolled out and evaluated.

## 1. Introduction

Homelessness is a global public health problem that comes with high social and economic costs for many communities [1,2]. It is a complex issue, defined as the lack of a “fixed, regular, and an adequate nighttime residence” [3] (p. 2). For young people experiencing or at risk of homelessness, including those youth who are street homeless or who live in unsafe, inadequate, or insecure housing, the challenges they experience are even more pronounced. Housing services provide safety and shelter to homeless youth in crisis and support them to address the significant and often co-existing mental and physical health issues that impact their integration into society [4]. Some of these issues include anxiety, depression, inadequate nutrition, sexually transmitted infections, substance abuse, and trauma and distress [4,5,6]. Further barriers include not being in employment, education, or training (NEET) and lacking opportunities to develop the life and relationship skills required for independent and fulfilling lives [5]. Indeed, the most recent “Young and Homeless” report by Homeless Link indicated that being NEET was one of the top 3 support needs of young people accessing homeless services in the UK, along with mental health challenges and a lack of independent living skills [5]. A key role played by housing services is to support young people experiencing homelessness in accessing educational and training opportunities as well as paid employment.

Past research highlights the need for interventions that enable young people to develop their strengths while also acquiring skills to manage their emotions and build positive relationships [6,7]. Strengths-based interventions recognize young people’s capacity for growth through exploring inner strengths and resources, empowering them with self-determination and personal efficacy [8,9]. While strengths-based approaches to case management have been reported [10,11,12], there is still a lack of strengths-based programs to engage young people experiencing homelessness in meaningful activities, as part of their healthy and optimal development toward independent adulthood (for a review, see [13]).

### 1.1. The My Strengths Training for Life™ Program

Inspired by the mental skills training interventions that are typically used in sports settings (e.g., [14]), the My Strengths Training for Life™ (MST4Life™) program was developed in collaboration with staff and young people from a large housing service operating in the West Midlands (UK) in a long-term community–academic partnership (>8 years) [15]. MST4Life™ is part of larger structural action to reduce the many health and social inequalities (e.g., lack of transportation, barriers to accessing mental health support, difficulties obtaining secure and long-term housing) as well as the powerlessness of young people experiencing homelessness in the face of group-based discrimination (e.g., the stigma of homelessness status, the excessive force from and discrimination by law enforcement) [5,15,16,17]. The Housing Service (i.e., the charity partner) initiated discussions with sport psychology researchers to explore innovative ways of engaging young service users and aid their transition into independent living, while also creating staff opportunities for continuing professional development.

In sport, a mental skill is defined as a regulatory capacity to maintain or develop a psychological outcome (e.g., self-confidence, resilience; [18]). Many of the psychological techniques used by athletes (e.g., goal-setting, planning, routines) are transferable to other contexts and could, therefore, support young people experiencing homelessness in achieving certain life goals [19]. As explained by Cumming et al. [15], many young people experiencing homelessness will have a history of adverse childhood experiences (ACEs) that can lead to difficulties in regulating emotions and coping with stressful events. Like athletes, they may benefit from systematically developing mental skills for improving intrapersonal qualities, such as confidence, resilience, and self-regulation, along with interpersonal qualities, such as being able to work in a group and show respect for others, to maximize their potential and cope with ongoing exposure to pressure and stressful situations. In other words, mental skills are important assets for young people and are fundamental for their optimal development, functioning, and health.

MST4Life™ is an experiential and strengths-based psychoeducational program intended to promote the use of mental skills in young people, to enable them to build resilience by better self-regulating their thoughts, feelings, and behaviors, and transferring the use of these skills to other life domains (e.g., education, employment, or training). The long-term outcomes of participating in the program are predicted to be: (a) reducing the likelihood that a young person will present as homeless; (b) improving mental, social, and physical health over the course of their life; and (c) lowering rates of mental illness and mortality (for the rationale, logic model, and description of MST4Life™, see Cumming et al. [15]).

In adopting a strengths-based approach, MST4Life™ capitalizes upon positive youth development (PYD; [20]) as the framework to inform intervention content and delivery. Within PYD, the complex and often intertwined problems experienced by young people are not ignored, nor are they the focus [21]. Instead, PYD programs aim to develop both internal and external assets through a skill-oriented lens; they promote thriving and growth by aligning young people’s strengths with positive, affective relationships with caring adults, challenging experiences, and skill-building opportunities offered in diverse activities and settings [22,23,24]. Through PYD, young people build the capacity to intentionally self-regulate, which in turn enables them to capitalize on opportunities within their environments to develop positive assets [21], boosting their resilience, wellbeing, and healthy development [25].

As a means of facilitating positive relationships between facilitators and young people, MST4Life™ is additionally framed by self-determination theory (SDT; [26]); a theory of motivation that argues that all individuals have an innate capacity for growth and development, provided that their basic psychological needs are met. According to SDT, autonomy, relatedness, and competence are basic psychological needs that are common to all individuals and essential for promoting optimal development, functioning, and health [26]. Autonomy refers to the need to feel volition and have a sense of choice in one’s actions, relatedness is the need to feel connected to others and integrated into a larger social whole, and competence is the need to feel efficacious and believe that one’s actions result in intended outcomes. MST4Life™ program facilitators support these needs [27,28,29] by creating a relaxed and enjoyable environment that offers autonomy support (e.g., regular opportunities to make personal choices and give input), interpersonal involvement (e.g., demonstrating acceptance, care, warmth, understanding, and respect for participants), and appropriate structure (e.g., the provision of clear instructions and guidance, positive expectancy, optimal challenges, and constructive feedback; [30]).

### 1.2. Study Aim

MST4Life™ was iteratively developed, delivered, and evaluated through action research cycles [15]. This study describes an important step that was undertaken after the formative work was completed (i.e., a narrative literature review, consultation via focus groups with key stakeholders, including young people) and before the main roll-out occurred [15]. Its aim was to establish the feasibility study of engaging young people experiencing homelessness, who are aged 16–24 years old and living in supported accommodation, in a multifaceted intervention with meaningful opportunities to recognize, apply, further develop, and transfer the use of their mental skills into different contexts. This study would help to ensure that the program is engaging [31] and rooted in evidence [8,20]. It was hypothesized that young people would be willing to engage with MST4Life™ and find its content and delivery style to be appealing. The study was designed to answer the question, “Can it work?” before evaluating “Does it work?” and “Will it work?” in later iterations [32]. The focus of this feasibility study was on evaluating the recruitment capability and plans, data collection procedures, acceptability, and suitability of MST4Life™, along with a preliminary evaluation of participants’ responses to the program [33].

The present study is part of a large community-based participatory research (CPBR) project [15]. As MST4Life™ was designed with substantial input from staff and service users, it seemed appropriate to test the fit of MST4Life™ within the constraints of a real-world setting, as opposed to using a highly controlled trial. The latter may reduce the external validity of the intervention [32]. A mixed-methods approach, aligned with a pragmatic methodological tradition, was used to understand the views of these key stakeholders (i.e., young people experiencing homelessness and their support workers). A concurrent nested design with mixing methods was used because a single type of data would not have been sufficient for addressing the study’s aim [34]. Both qualitative and quantitative data were collected but were used for different purposes. The qualitative data was collected to better understand the appropriateness of the recruitment and evaluation methods, as well as the acceptability of, and reactions to, MST4Life™ from the perspectives of both young people and housing service staff. The quantitative data was used to supplement this information by describing young people’s attendance of the program.

## 2. Materials and Methods

### 2.1. Program Recruitment

Recruitment to pilot the MST4Life™ program was carried out by two support workers, who approached young people directly and invited them to take part. The support workers also encouraged young people’s participation throughout the program by reminding them about sessions and arranging transportation when needed. Young people were invited to participate in the pilot MST4Life™ program if they: (a) were experiencing homelessness; (b) lived in one of the housing services’ long-term supported accommodation sites (i.e., where young people typically live for between 6 and 12 months); and (c) were referred by their support worker due to their limited engagement with education, employment, or training opportunities (e.g., were currently NEET or at risk of dropping out of education, employment, or training opportunities) and/or perceived lack of mental skills (e.g., found it difficult to set goals or make plans).

### 2.2. Program Description

The pilot MST4Life™ program was delivered face-to-face to participants in groups over two phases: (1) 8 weekly sessions of 1.5 to 4 h in length (April–June 2014); and (2) a short (4-day/3-night) residential course at an outdoor pursuits center in the Lake District, UK (September 2014). A residential course is defined as one when participants stay in a residence during the course; in this case, the outdoor pursuits center provided participants with accommodation (i.e., single-sex shared dormitory-style bedrooms for participants and separate bedrooms for visiting staff, single-sex shower rooms with toilets, kitchen/dining room, and lounge). The first phase (community-based) targeted the initial development of both existing and new mental skills (e.g., goal-setting, planning, using strengths) (Table 1).

The weekly sessions in this phase were jointly led by two researchers with experience in delivering either mental skills training programs in sports settings or experiential learning in outdoor pursuits settings. Facilitators focused on promoting a relaxed and needs-supportive environment aligned with SDT [26]. Young people were invited to give input and be involved with decisions throughout the program (e.g., the time/day for sessions to take place, the time needed to complete activities, choice of refreshments, selection of outdoor adventure activities during the residential course), and were encouraged to take ownership of their behaviors and emotional management (i.e., autonomy-supportive). Another salient feature was the emphasis on developing relationships with the young people through rapport-building, validation, empathic listening, and open questions (i.e., involvement). Finally, activities were designed using an experiential learning approach [35] to be fun, accessible, and appealing to young people, and comprise increasing levels of challenge (i.e., structure). To foster the transfer of learning beyond the sessions, young people were also encouraged to notice when opportunities were presented outside of sessions to try out new ideas and skills (e.g., seeking employment; [36]).

The second phase of MST4Life™ was a residential trip to the University of Birmingham’s outdoor adventure center. This center, located in the Lake District, UK, was chosen owing to its connections with the university and the broad range of facilities offered. Its location was crucial to the undertaking of Phase 2; this trip away from the young people’s normal urban environment aimed to promote and further develop their mental skills in a novel, fun, and challenging setting for the transfer of learning [28]. The activities (e.g., high ropes challenge, mountaineering, zip wire) on the residential trip were led by qualified instructors and the necessary safety equipment was provided by the outdoor adventure center. MST4Life™ facilitators remained present to provide consistency in the relationships already developed and specific support on the use of different mental skills during the activities. The second phase provides young people with opportunities for learning through direct experience, receiving feedback, and reflecting on their experiences. It was also specifically tailored to provide challenging opportunities for young people to transfer and further enhance the mental skills that they developed in the first phase [28,36].

The young people who took part in the pilot MST4Life™ program received no external incentive for their involvement in the program or its evaluation. However, refreshments were provided during sessions.

### 2.3. Study Design, Setting, and Procedures

The feasibility study followed a non-randomized design consisting of a single group of participants at two different intervention sites in Phase 1 and a combined group in Phase 2 at a single intervention site. In Phase 1, data was collected both during and six weeks after the program finished; in Phase 2, data were collected on the first three days (Figure 1). For their data to be included in the study, which was determined retrospectively after the program took place, participants had to have attended at least one session of MST4Life™ and provided their informed consent for their data to be included in the evaluation. All young people who took part in the pilot MST4Life™ program met these criteria and their data were therefore included in the feasibility study. Contributing to this evaluation was optional and the refusal to provide data did not restrict young people from participating in the program or the broader support offered by the housing service.

Two different sites were chosen by the housing service to host the pilot MST4Life™ program because they accommodate and support the highest number of young people classified as NEET and those with the most complex needs, thereby providing a realistic test of whether it was possible to engage young people experiencing homelessness with MST4Life™ in this setting. Young people were permitted to join the program after it had begun.

### 2.4. Data Sources

Data for the study was obtained via attendance records and qualitative methods during and after each phase of MST4Life™ (i.e., diary rooms and focus groups).

#### 2.4.1. Diary Rooms

A semi-structured video diary room was employed to gain reflections from the young people. A semi-structured diary room can aid participants’ learning as a self-reflection tool. The diary room provided participants with a private yet engaging means to reflect and give feedback on the MST4Life™ program content and delivery [37]. The approach is flexible insofar as participants can direct the content and duration of their responses (i.e., what and how much they say about a topic) to pre-determined questions and topics of interest (see Table 2).

Most participants provided individual video-recorded entries, but they could also do so in pairs and/or audio-record if they felt more comfortable with that. Entries were recorded using a Canon 60D camera, with questions provided on cards. In Phase 1, young people were invited to give diary-room entries at two time points (i.e., sessions 4 and 7). Four young people volunteered to participate (44.44%), with reasons for non-participation being due to non-attendance at the session where the diary room was present, or young people expressing that they were not comfortable with having their views recorded. During the residential trip in Phase 2, the diary room was more readily available (i.e., days 1–3) and was participated in by 9 young people (90%). Diary-room entries ranged from 1 min 26 s to 10 min 49 s (Mean = 4 min 49 s).

#### 2.4.2. Focus Groups

Three focus groups were held 6 weeks after Phase 1 of MST4Life™, with separate discussions for young people (two focus groups, 5 participants in total) and support workers (2 participants). The semi-structured topic guide was phrased in the form of open questions, allowing for both positive and negative experiences to be discussed, and participants were encouraged to be as honest as possible. The facilitator also welcomed negative feedback about the program and explained that it was an opportunity for the young people to influence its future improvement. The focus groups lasted an average of 46.33 min and were audio-recorded and transcribed verbatim.

### 2.5. Methodological Position

We (the researchers) locate the overall MST4Life™ project within a pragmatist paradigm, with explicit reference to Dewey’s [38] philosophical tradition and instrumental view on knowledge. Dewey asserts that knowledge is created through action and the reflection thereof; in turn, knowledge is the basis of action and is used to inform change and improvement. As such, Deweyan pragmatism aligns well with action research as the chosen method of inquiry, the active role of researchers in promoting change, and experiential learning as the adopted pedagogical approach for the intervention [39]. This position additionally guided the research toward selecting methods that could be changed, adapted, and intelligently used to provide information for the evaluation of MST4Life™ and its future improvement [40]. This study considers young people’s lived experiences of homelessness as a source of knowledge that can feed directly back into the knowledge base around how to develop a feasible, accessible, and engaging program.

### 2.6. Data Analysis

Analysis of the verbatim transcripts stemming from diary-room entries and focus group discussions was conducted using NVivo (Version 10.2) and followed the six-phase process outlined by Clarke et al. [41] for interpretive thematic analysis (TA). This method focuses on the participant’s standpoint, including how they experience and make sense of the world [42]; in this case, the MST4Life™ program within the context of living in supported accommodation. The six phases of TA comprise data familiarization, initial coding, the sorting of codes into themes, reviewing of themes, naming and defining themes, and producing a report [41]. TA is appropriate for this investigation, given the large size and diverse nature of the obtained data and the research goal of identifying, analyzing, and reporting patterns across this data set in relation to the research question: what is the feasibility of engaging young people experiencing homelessness in the MST4Life™ program? An inductive approach was used to gain a deeper understanding of the data.

Within Deweyan pragmatism, a broad view of credibility is adopted that extends beyond methodological rigor to emphasize the role of reflection in gaining a new understanding of beliefs, problems, and prior knowledge [40]. In this study, reflection was infused throughout the data analysis, as well as in the additional steps taken to establish credibility. The researchers regularly met to discuss the program implementation and evaluation; they engaged in collaborative reflexivity [43] to question any assumptions and acknowledge personal biases and inclinations toward believing that young people experiencing homelessness would enjoy participating in a strengths-based program designed to improve mental skills [10]. Other strategies included prolonged observation, peer debriefing, and providing a thick description of the results using quotes to represent the different viewpoints (e.g., young people vs. staff) to encourage the reader’s own reflections. Moreover, the reflections of the different stakeholders were solicited during the analytical process in separate meetings with frontline and management staff and the Board of Directors of the housing service. These discussions were viewed as additional opportunities to clarify, strengthen, and gain additional thoughts about the data [44].

## 3. Results

### 3.1. Recruitment

Target recruitment size was between 6 and 12 participants per group, so as to be maximally engaging for young people. Of the 78 potential participants living at the two supported accommodation sites, 15 people (M age = 19.99 years, SD = 2.42; 60% male, 40% female) enrolled in at least one intervention phase (both phases = 4; Phase 1 only = 5, Phase 2 only = 6), representing a reach of 19.2% of potential participants. Due to the dynamic nature of supported accommodation, four young people had successfully moved into independent accommodation by Phase 2 and were either unable to attend the residential course due to new education, employment, and training (EET) commitments or were no longer in contact with their support worker. For this reason, an additional 6 participants were recruited for Phase 2. At the time of their participation, the young people had lived in supported accommodation for an average of 1.6 years (SD = 1.0) and 67% were NEET. Participants were of various ethnicities and 27% spoke English as a second language (Table 3).

A high level of commitment was required from the support staff during recruitment and throughout the program to support attendance. Young people appreciated the role played by support workers in making them aware of the program. However, they believed that improvements could be made to the recruitment plan by advertising the program via posters, timetables, and leaflets. One young person believed that better advertising would lead to more young people taking part: “If they advertised it a lot more then people would attend. I reckon they would, man [believe they would]. You need to get the advertising across”.

### 3.2. Appropriateness of Evaluation Methods

Young people displayed an initial reluctance to be involved in qualitative data collection, suggesting that more time was required to establish trust and rapport. The more confident young people gave short diary-room entries in Phase 1, and five participated in the focus group discussions that took place between phases. By Phase 2, nearly all participants had completed a diary-room entry, including those who had not participated in Phase 1. The young people appeared more relaxed and gave longer and more in-depth entries later in the program. The average diary-room entry was 2 min 36 s in Session 4 of Phase 1 (Mean = 26 s per question) and lengthened to 6 min 43 s by Day 3 of the residential course in Phase 2 (Mean = 45 s per question).

Another notable feature was that the diary room served to encourage young people to engage in reflective learning and provided meaningful opportunities for developing communication skills. For example, one participant described how they would previously have been hesitant and unsure of what to say when being filmed, “Like, just me now speaking to the camera, before I know I would have been like, ‘if’ing’ and ‘er’ing’ [describing themselves as being unsure of what to say], don’t know what to say, this and that, but it’s become a lot better, I think”. On occasions, however, some of the young people found it hard to describe their experiences, with another participant saying, “I just feel some changes in me, I can’t really explain how I feel, it’s kind of hard as well, but I just feel something in me has changed, I don’t know what”.

### 3.3. Acceptability of the Program

#### 3.3.1. Attendance

Young people’s pattern of attendance throughout the program indicated that one session per week over 8 weeks in Phase 1 and 4 days/3 nights of residential course in Phase 2 was appropriate, in terms of both structure and length. Of the 9 young people who participated in Phase 1, the attendance rate for each session was 75% (see Figure 2). All 10 young people participating in the residential course completed the activities and remained at the outdoor adventure center for the entire duration of the trip (i.e., 100% attendance).

The high level of attendance in Phase 1 for a weekly program was considered an important feature to the young people because it meant, as one young person put it, that “there was a lot of people you could, like, connect with and talk to”. Reasons for non-attendance included attending a job interview, visiting family in another city, and a dispute occurring between two members of the group outside of MST4Life™, which led to one participant declining further involvement beyond Week 7. The level of attendance was deemed highly successful by the support workers, who commented: “To have a program running for 8 weeks and have them engaged right the way through, you gotta [have to] be doing something right”. Indeed, young people expressed their willingness to engage with an even longer Phase 1 and/or repeated sessions to enable their further development. A young person explained: “It’s good when you do things more than once because you just feel greater and greater every time you do it”.

Staff favorably compared attendance at MST4Life™ to other training courses offered by the housing service. A support worker explained: “They don’t turn up for nothing. Even a 2-week course they’d probably do half of the first week. But they’ve shown that they can be quite consistent [with MST4Life™]”. A staff member also described: “What was interesting was that the only incentive, right, was for their personal development”.

As is mainly relevant in Phase 1, contextual features, such as the support workers and the location of the sessions, contributed to ongoing attendance. The support workers initially provided encouragement to the young people. They noticed, however, a decreased need to prompt young people to attend each session as the program progressed, indicating improved self-motivation through the program, with one explaining how they would initially have to knock on the door of each young person:

“*At the start it was a constant—’bam, bam, bam’ [knock on the door], until it got to the stage where I would remind them [the day before] and give them a knock in the morning. Simple as that. Some of them I wouldn’t even have to, they’d be ready.*”

The proximity of sessions to the young people’s accommodation was also considered to be vital for maximizing attendance in Phase 1. A young person described how convenient it was to attend sessions by saying: “We live in the same place … you’re not going to miss nothing innit [are you], so if you miss something then it’s your fault”. Similarly, possessing an initial level of confidence was also deemed important. A young person explained:

“*They [young people] need the mentality and the confidence as well, like if you ain’t [are not] confident to do it [take part in MST4Life™] there’s no point, like, setting yourself up and looking like an idiot. So yeah, a lot of people need to be a little more confident when they are doing this.*”

#### 3.3.2. Engagement

Engagement during sessions was evident by the young people having fun while still learning. A young person described how they felt drawn into MST4Life™ from the outset of the program and the enjoyment that they experienced during the sessions helped to maintain their engagement:

“*Just from that one week it drawed [sic] me in anyway. It drawed me [describing themselves being attracted to the program] in for the rest of the weeks because I had a good time and it was fun. And I was learning as well as having fun. I loved it.*”

Indeed, the participants would become engrossed in learning, with one young person saying that learning occurred “without even realizing it”. Others explained that during the sessions, they often didn’t “pay attention to the time length” of the sessions, and instead they focused on “just getting the job done”. A support worker explained how MST4Life™ “didn’t overdo the paperwork side of things, you tried to keep it as practical as possible, it kept them fully focused and engaged with what’s coming, ‘what’s next?’”. In turn, the participants appreciated the relaxed nature of the program; as one young person put it: “It’s not too pushy and it’s not too planned. So you just go with the flow. When there’s things that are planned, there’s not really that much fun”.

### 3.4. Participants’ Reactions to the Program

When reflecting on their experiences, many of the young people expressed surprise at their initial low expectations of the program having been surpassed. They felt a sense of pride over what they had achieved. A young person explained how they had no preconceived expectations of how they would benefit from taking part: “We never actually thought we would gain nothing from the course. But as the course went through … we started to realize the changes … I’m just proud of it”. They did not want the program to end and expressed gratitude for having had the opportunity to take part. They also commented on how beneficial it would be for other young people. As another young person observed:

“*Not a lot of people can do this [MST4Life™ program] but now maybe the way this group that we’ve got now can improve and get better and make doors open for other people that want to go on it.*”

Similar positive reactions were additionally expressed for the approach taken by the facilitators. Few direct references were made to autonomy-supportive strategies (e.g., giving input and being invited to make decisions) but young people conveyed acceptance of the structure (e.g., the challenge level was well-balanced and ensured young people were pushed to use and develop a range of skills). The interpersonal involvement provided by facilitators was also appealing; they were described by the young people as “upfront and straight”, “respectful and genuine”, “really friendly”, and “attentive and supportive”. It was evident that young people perceived themselves as having a great rapport with the facilitators. Examples of the positive feedback included statements such as “you’re doing a really good job for everyone” and “I love the staff”. The support staff also agreed: “The young people took to you guys very well”, which in part was due to the facilitators being young-person-led. One support worker explained:

“*You’ve met them, met them on their terms… And if ever there was a week where someone felt they wasn’t [sic] feeling it this week you wasn’t in their head saying ‘Come on, come on!’… You were like ‘No problem, do what you’re doing’, and I think that worked for them.*”

## 4. Discussion

The aim of this study was to report on the feasibility of a PYD program for young people experiencing homelessness, MST4Life™, as determined by participant attendance, engagement, and reaction. Following the recommendations of Bowen et al. [32] and Orsmond and Cohn [33], the focus was on answering “Can it work?” by evaluating the recruitment capability and plans, data collection procedures, acceptability and suitability of the program, and a preliminary evaluation of participant responses to the program. Overall, the findings indicated that the program shows promise of being feasible for this population, as evidenced by young people’s views that the evaluation methods were appealing, the delivery style of the facilitators was engaging, and that the autonomy- and relatedness-supportive climate of the program was successfully cultivated.

MST4Life™ addresses a gap in strengths-based psychoeducational programs aimed at promoting healthy and optimal development towards independence and adulthood in vulnerable older adolescents/emerging adults. The program was developed in collaboration with the staff and service users of the housing service. Together, we worked in partnership to identify the specific program goals and priorities, the nature of the activities, and the intended atmosphere of the program. It is likely that having this input from the outset contributed to the necessary “buy-in” from staff that led them to encourage young people to take part, thereby leading to those young people perceiving the content of the program to be enjoyable, relevant, and meaningful to them. In the future, the recruitment plan will combine this approach with other methods for advertising the program, based on the young people’s suggestions. That it was the young people who identified a need for better advertising reinforces the benefit of the collaborative approach in this study. In planned future collaborative research around MST4Life™, it will be possible to explore the ways in which essential stakeholder feedback can be integrated into the program.

### 4.1. Evaluation Methods

Aligning with the researchers’ methodological position of Dewey pragmatism, the program was refined through continual reflections on whether the data collection methods (e.g., attendance, semi-structured diary rooms, and focus groups) were enabling the researchers to: (a) understand what made MST4Life™ acceptable and attractive to its participants; (b) test these assumptions with the stakeholders by including both staff and young people in the evaluation; and (c) be critical about what areas could be changed to improve program uptake and adherence. The results suggest that major changes to the evaluation model were not needed prior to larger-scale implementation; however, further improvements were planned. For example, alternative methods of collecting information at baseline were considered to accommodate young people who are not comfortable with being audio- or video-recorded (e.g., questionnaires). A reluctance to engage in qualitative research without first having established trust in and rapport with the researchers is consistent with findings from other studies [14,45].

Using questionnaires in the future may help to overcome problems with personal expression as part of a flexible strategy for data collection, thereby avoiding the exclusion of young people from the evaluation due to its methods and ensuring that the findings are more representative of the participants [46]. Adopting a pragmatic stance encouraged sensitivity toward the context in which the intervention was delivered and evaluated, along with reflection on how this context may have interacted with participants’ experiences [40], such as practical support by support workers (e.g., knocking on doors to remind participants when sessions were running). This reflection reinforces the importance of a “wrap-around approach” and the need to understand the environmental factors influencing the research, before wider scale implementation and testing, and indeed before any change is made to how services operate [47].

### 4.2. Program Acceptability

Attendance indicated feasibility within the present study and the 75% attendance in Phase 1 was favorably compared by staff from the supported housing service to other programs. Variation in attendance was previously noted as an issue in a review of interventions for homeless children and young people that were similarly aimed at promoting inclusion and reintegration [48]. However, the attendance figures are similar to those reported for a social enterprise intervention involving homeless youth, reporting an 80% attendance rate [49]. To maximize the effectiveness of the intervention when it is scaled up [50] and to understand what might contribute to attendance in the present study, the analysis focused on identifying what contextual factors might be contributing to attendance on the MST4Life™ program. The three main factors identified were lack of external incentive, staff support, and individual characteristics.

The lack of external incentives in MST4Life™ indicated that attendance was intrinsically motivated. In contrast, staff reported lower uptake by young people in the present study to other initiatives that offered financial rewards for participation. Rotheram-Borus et al. [51] similarly reported high engagement of young people experiencing homelessness in research that did not involve incentives, whereas Coren et al. [48] reported insufficient evidence for their review to draw conclusions about the impact of incentives on participation. SDT, however, would suggest that such payment may undermine more autonomous forms of motivation [26]. Therefore, we recommend that researchers consider the effect of including financial incentives as attendance motivation. In this study, the lack of financial incentive points to the success of the program in terms of appealing to young people accessing housing support. Indeed, Parry et al. [28] found that meeting basic psychological needs was a key factor contributing to young people engaging with the program.

The finding of staff support as important for program uptake is consistent with PYD research. Newman et al. [52] found that the staff play a central role in delivering skills training within a community-based setting and that their input facilitates the transfer of learning. Kurtz et al. [53] similarly highlight the vital role that support staff play in helping young people experiencing homelessness to build trust, foster caring relationships, be held accountable and thereby develop autonomy, as well as providing practical support, advice, and guidance. Our findings reiterate the crucial role played by support workers in supporting young people’s development within housing services.

Trust is related to the third factor affecting program uptake and attendance, namely, the characteristics of the young people. Introducing a new program into the supported accommodation site brought what staff described as a “fear factor” about what participation would involve. The program seemed to attract those individuals who had enough confidence and enthusiasm to overcome this initial fear: a finding that suggests baseline differences may exist between those who agree to participate in MST4Life™ and those who decline [54]. In this study, participants reported being curious, open-minded, and willing to prioritize the program over other pursuits. Even in a group typically described as “hard-to-reach”, there will be individuals who are psychosocially better equipped to engage than others (e.g., social competence; [55]). Kurtz et al. [53] also describe how the readiness to engage, often based on perceived trustworthiness, is an important factor influencing engagement.

Reflecting the dynamic nature of supported accommodation, only four out of 15 young people participated in both phases of the program. Rather than viewing this as a limitation, a pragmatic decision was made to use this unexpected opportunity to explore whether Phase 1 was a necessary pre-requisite for Phase 2 [56]. Facilitators observed that those young people who participated in Phase 1 were better able to adapt to the novel outdoor adventure challenges provided in Phase 2, suggesting that young people benefitted more from the residential course if they had previously completed the eight weekly sessions of the community phase [26]. When possible, in line with seasonal constraints (i.e., the outdoor pursuits center is closed over the winter months), a shorter gap between phases is needed to improve the feasibility of the two-phase delivery model and reduce the potential for a drop-off in participation.

Experiential learning, as the chosen pedagogical approach, was a key contributing factor to engagement in the program [35]. In keeping with PYD, this theme suggests that young people were autonomously motivated to take on the challenges posed within the program and enjoyed having control over their own learning process; that is, acting as constructive agents in their own engagement [57]. In order to maximize engagement, this study has demonstrated that trust is an important factor [58]; this reinforces the need to encourage a supportive, relaxed [59], and psychologically informed environment (PIE; [60]), where facilitators play a key role and are young-person-led. Certainly, positive and mutually beneficial relationships are critical to adolescent development, providing opportunities for emotional connection and attachment, and are also central to achieving effective positive youth development by opening up new networks and resources [25]. In this study, facilitators aimed to create a program atmosphere that supported the basic psychological needs of the participants by using the motivational strategies of being autonomy-supportive, providing structure, and encouraging interpersonal involvement [29,30]. Collectively, the strategies for promoting psychological needs appear to be core elements of the program for inclusion in future implementation. From a theoretical perspective, young people are indeed more likely to experience psychological growth and well-being when their psychological needs are satisfied [10,26]. SDT is proposed as a way to underpin PIE by providing a psychologically and theoretically informed strengths-based framework to guide the development of targeted programs for young people experiencing homelessness.

A rich array of positive reactions to the program was evident within the data, which suggested that MST4Life™ was both acceptable to the participants and fulfilling a need among young people in their situation. In line with the expressions of gratitude from young people in this study, it will be necessary to explore the ways in which the program could be scaled up and out, so that others may be offered access to the same potential benefits by involving key stakeholders of staff and young people in a process of collaborative knowledge translation research. This will be important not only in supporting young people more fully, in addition to “wrap-around” support, but also, crucially, in terms of supporting individuals to go on to lead independent adult lives beyond housing support.

### 4.3. Limitations and Next Steps

The main limitations of this study were the small sample size and the use of a cohort design rather than randomization into intervention vs. standard care/wait-list control groups. Two main reasons guided the design choice: (a) the co-produced nature of the program, and (b) the desire to test the intervention in the real world under the practical constraints of that setting [54]. While this may be viewed as a limitation, the nature of community-based research does not necessitate the use of a randomized trial, instead highlighting the inherent issues of this typically clinical approach [61,62]. Given that this feasibility study was conducted at two different sites within a single housing service, further implementation has since occurred at more long-term accommodation sites, as well as in shorter-stay accommodation sites (e.g., 30 days or less) within the same housing service [15,27,28,29,54,55]. Between 2015 and 2020, more than 600 young people took part in MST4Life™ and evidence is accumulating to demonstrate the short- and long-term impact of the program, including the finding that MST4Life™ participants are two times more likely to transition into EET and independent living, compared to standard care by the Housing Service [63]. Future research is also needed to test MST4Life™ within other similar services for young people experiencing homelessness or who are at risk, such as young people with a history of ACEs, those who have been excluded from school and/or leaving care, young offenders or justice-involved youth, and NEET young people with mental health difficulties [15]. However, the naturalistic setting in which the program was conducted makes it more likely that the lessons learned will be transferable to other homeless young people or those at high risk of homelessness.

## 5. Conclusions

This study has demonstrated the need within housing services for a strengths-based psychoeducation program that is community-informed and framed by a PYD approach. MST4Life™ uniquely extends the reach of mental skills training beyond a sport psychology context, to support young people who are homeless and moving toward education, employment, and training. It additionally meets the long-standing call for programs within this population to take a strengths-based approach, enabling young people experiencing homelessness to recognize and build upon their existing assets and resources, as opposed to focusing on their deficits and problems. Moreover, the content and delivery of this program are distinctive because of its explicit links to both psychological (e.g., self-determination theory) and pedagogical (e.g., experiential learning) theories. That is, the program is designed to encourage young people to reflect upon and learn from the fun and meaningful experiences offered across the two phases of the program, in a way that facilitates their basic psychological needs for autonomy, competence, and relatedness as well as to transfer this learning to other life domains. By doing so, MST4Life™ offers the potential for a new model of intervention for services that support young people experiencing homelessness or those at risk. This study and the extant literature on MST4Life™ [15,25,26,27,51,52] also illustrates how working in equal and mutually beneficial partnerships with the community can contribute to the development of new intervention approaches that more effectively understand and respond to the inequalities experienced by socially disadvantaged communities.

## Figures and Tables

**Figure 1 ijerph-19-03320-f001:**
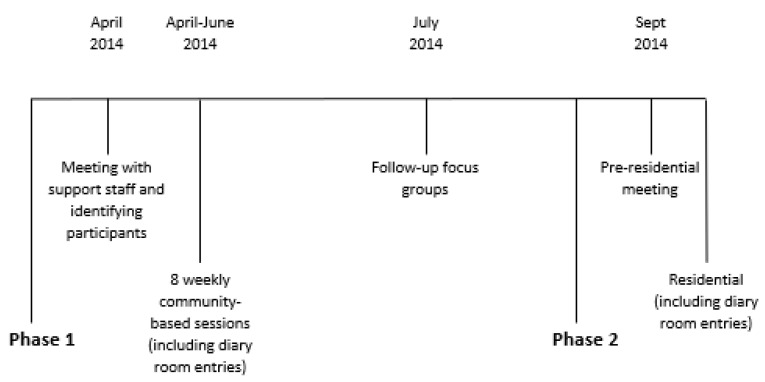
Intervention Timeline.

**Figure 2 ijerph-19-03320-f002:**
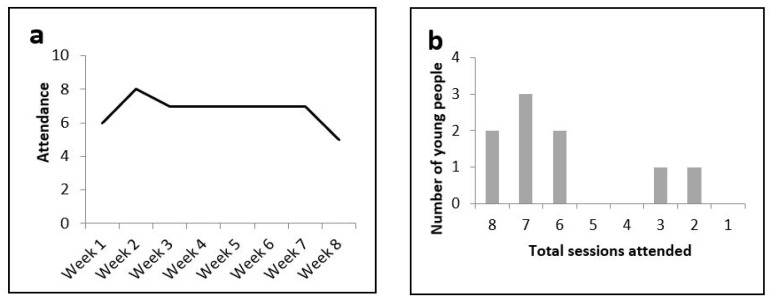
(**a**) Attendance record during Phase 1 of MST4Life; (**b**) the total number of sessions attended by each young person.

**Table 1 ijerph-19-03320-t001:** MST4Life™ program content and learning objectives.

Week	Name	~Duration	Learning Objective
1	Introduction and Marble Run Challenge	1.5 h	Problem-solving and interpersonal skills
2	Ideal Selves and Strengths Profiling	2 h	Aspirations, self-confidence, and intrinsic motivation
3	Photo Safari	3 h	Interpersonal skills, social competence, and organizational skills
4	Apprentice Master Chef-Planning	2 h	Organizational skills
5	Apprentice Master Chef	4 h	Self-regulation, organizational skills, interpersonal skills, self-confidence, and aspirations
6	Dream Team and White Knight Challenge	2 h	Coping skills, self-regulation, and responsibility
7	Warm Fuzzies and Dragons De-Planning	2 h	Self-confidence and social competence
8	Dragons’ Den and Preparation for the Residential Course	2 h	Self-regulation, organizational skills, interpersonal skills, and self-confidence

Note: ~ indicates the average session.

**Table 2 ijerph-19-03320-t002:** Diary room questions.

Time Point	Questions
Phase 1 (Week 4)	What were your reasons for getting involved in this program?
	2. What did you expect this program to be about?
	3. What did you hope to get out of attending this program?
	4. What have you most enjoyed about the program?
	5. What have you least enjoyed about the program?
Phase 1 (Week 7)	What have you found most challenging during this program?
	2. Did you manage to overcome this challenge? How?
	3. What are your views on the style and approach of the people who delivered this program?
	4. Have you got anything out of attending this program? If so, what?
	5. Are there any improvements we could make to the program? If so, what?
Phase 2 (Day 1)	How do you feel about being on this outdoor adventure course?
	2. What do you hope to get out of being here?
	3. What has been the best and worst thing about Day 1?
	4. Have you learned anything so far? If so, what?
	5. Do you think any of the skills you developed during the mental skills training will help while you are here? Please explain.
Phase 2 (Day 2)	How have you found the outdoor adventure course so far?
	2. What do you think of the staff and accommodation facilities here at the center?
	3. What have been your best and worst experiences so far?
	4. Has your mindset/state of mind changed in any way during the course?
	5. Is there any additional information we could have given you before you came that would have helped you prepare for the course?
Phase 2 (Day 3)	What have been your most memorable experiences of the outdoor adventure course?
	2. Have you noticed any changes in the way you face challenges and/or control your emotions?
	3. Has the outdoor adventure course changed the way you view yourself in any way?
	4. Will you do anything differently after having completed this course?
	5. Do you have any ideas for how we could improve the course for young people in the future?

Note: Phase 1 was the 8-week community-based program and Phase 2 was the 3-night outdoor adventure residential course.

**Table 3 ijerph-19-03320-t003:** Demographic characteristics of the study sample.

Characteristic	Study Sample % (*n*)
Gender	
Male	60 (9)
Female	40 (6)
Country of birth	
UK	66.7 (10)
Somalia	13.3 (2)
Jamaica	6.7 (1)
Portugal	6.7 (1)
Sweden	6.7 (1)
Ethnic group	
Black (e.g., African, Caribbean, or Black British)	46.7 (7)
White (e.g., British, Irish, Gypsy, Traveler, or other White backgrounds)	33.3 (5)
Mixed (i.e., multiple ethnic groups)	20 (3)
First language	
English	73.3 (11)
Patois	6.7 (1)
Portuguese	6.7 (1)
Somali	6.7 (1)
Swahili	6.7 (1)
Employment status	
Not in education, employment, or training (NEET)	66.7 (10)
Full-time education	20 (3)
Apprenticeship	6.7 (1)
Working part-time	6.7 (1)

## Data Availability

Data not Available. Due to the sensitive nature of the questions asked in this study, participants were assured that the raw data would remain confidential and would not be shared.

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
