# Peer review of "A Feasibility Study of the My Strengths Training for Life™ (MST4Life™) Program for Young People Experiencing Homelessness"

_ijerph, 2022, doi:10.3390/ijerph19063320_

Round 1

Reviewer 1 Report

Overall, this as an important and relevant piece of research. The inclusion of support workers and youth as participants improves the study's credibility. The combination of theoretical approaches is appropriate for this context, as is the thematic analysis for the method. However, there is not much written in the manuscript that supports it as a mixed methods project. This requires attention vis-s-vis quantitative data. As well, given the small sample size caution is required moving the program into a larger context. One of the limitations is the lack of comparison; perhaps another larger pilot study is in order?

 There are a few places where wording can be improved, and a few places that require elaboration:

-Define homelessness, mental and physical health issues

-line 36 replace not being employed with unemployed

-lines 51-54 provide some details re: inequalities/powerlessness

-lin2 62 define mental skills

-line 115 replace relevance with validity here and throughout manuscript

-line131 replace residential with residency here and throughout

-line 270 the quote seems lost, with no speaker, introduce the quote with the participant, here and throughout

As it stands, this work is publishable, but attention to the comments above will aid its relevance and validity. It appears that the program can work, but the more clarity and validity you can provide, the better it will be accepted and supported in practice.

Good luck.

Reviewer 2 Report

This is an important article that analyses validity of an intervention in NEET population. Even so, I think it may require some revision before it is acceptable.

  1. The focus of the intervention is not clear. The authors insist homelessness, but the intervention seem to have focused on NEET rather than homelessness, though the target population is young people who experienced homelessness. To make this point clear, the authors may articulate the relationship between NEET and homelessness, and then describe why it is especially important to promote mental skills of homeless youngers.
  2.   'Recruitment' section might be better be included as 'Materials and Methods'. In addition, please describe how the authors recruited participant, not just about data inclusion criteria. e.g. by advertisement, with/without rewards, etc. Also in the 'study design', they described inclusion criteria (b) as 'engaged in at least one session of the MST4Life program' . In this case, this is a retrospective study using pre-existing data. Is this a retrospective cohort, or did this recruit participant prospectively? Please make this point clearer.
  3. If this research was conducted retrospectively, was informed consent obtained from all the participants of the MST4Life, or did the selected participants alone agreed with the data usage? Please explain this point in the last part or in the methods.
  4. The references of the participants includes many dialects, which may not understandable for non-native readers. For example 'I reckon they would man', ' ain't', 'it drawed me'. I agree it is important to retain the participants original words to some extent, but some notes should be inserted so that non-native English researchers can understand the situation.

Round 2

Reviewer 2 Report

I think the authors have revised the manuscript well.